# An Automated Machine Learning Approach for Real-Time Fault Detection and Diagnosis

**DOI:** 10.3390/s22166138

**Published:** 2022-08-17

**Authors:** Denis Leite, Aldonso Martins, Diego Rativa, Joao F. L. De Oliveira, Alexandre M. A. Maciel

**Affiliations:** 1Mekatronik I.C. Automacao Ltda, R. Itapeva, 43a-Imbiribeira, Recife 51180-320, Brazil; 2Institute of Technological Innovation, University of Pernambuco, R. Benfica, 455-Madalena, Recife 50670-90, Brazil; 3Stellantis, Rodovia BR 101 Norte, Km 13-15a, Goiana 32530-900, Brazil

**Keywords:** discrete-event systems, fault detection, fault diagnosis, intelligent manufacturing systems, machine learning, smart manufacturing

## Abstract

This work presents a novel Automated Machine Learning (AutoML) approach for Real-Time Fault Detection and Diagnosis (RT-FDD). The approach’s particular characteristics are: it uses only data that are commonly available in industrial automation systems; it automates all ML processes without human intervention; a non-ML expert can deploy it; and it considers the behavior of cyclic sequential machines, combining discrete timed events and continuous variables as features. The capacity for fault detection is analyzed in two case studies, using data from a 3D machine simulation system with faulty and non-faulty conditions. The enhancement of the RT-FDD performance when the proposed approach is applied is proved with the Feature Importance, Confusion Matrix, and F1 Score analysis, reaching mean values of 85% and 100% in each case study. Finally, considering that faults are rare events, the sensitivity of the models to the number of faulty samples is analyzed.

## 1. Introduction

Industry 4.0 uses IoT, digital twin systems, and predictive maintenance technologies to improve business processes and gain a competitive advantage. In that scenario, Real-Time Fault Detection and Diagnosis (RT-FDD) is fundamental for increasing the reliability of production systems by preventing breakdowns [1]. The real-time aspect enables early interventions when abnormalities are detected, while the diagnosis feature supports precise maintenance actions.

It turns out that industrial processes may be continuous or discrete, and different RT-FDD approaches are required for each one since their behavior are distinct. Continuous processes, such as oil refineries and distillation systems, deliver products at a specific rate (e.g., ton/h, L/min) with uninterrupted operation. While discrete processes, such as Discrete Manufacturing Machines (DMMs), are executed in well-defined sequences of steps, with specific duration, delivering unitary items (e.g., bottles, boxes) [2]. Therefore, performing RT-FDD in continuous processes comprises comprehending the normal behavior of variables along time, in steady states and transitions, to differentiate it from anomalous or faulty behaviors. In contrast, in discrete systems, RT-FDD comprises understanding sequential operations, their duration, and how continuous variables behave in each operation.

In this context, this study focuses on performing RT-FDD in DMMs, which are widely present in the manufacturing industry. Their automation systems generally include knowledge-based (KB) fault diagnostic features that monitor sensors and parameters, and trigger alarms [3]. Furthermore, their behavior is typically described by a sequence of events, represented by inputs and outputs (IOs) [4,5], and may also involve continuous variables such as pressures, flows, temperatures, levels, positions, energy quality, power, or consumption [3]. Consequently, KB fault diagnostic is limited to a few known situations that can be humanly implemented due to the complexity and variety of the data types and the number of devices to be monitored in a DMM. Given that, manufacturing performance may be improved by overcoming the limitations of the KB solutions and delivering more and better diagnostics that support maintenance interventions.

Alternatives to KB systems are Physical Models (PM) and Data-Driven techniques such as Machine Learning (ML). However, while KB and PM approaches demand high engineering effort and deep knowledge about the machine’s behavior [6,7], ML does not, since it learns from machines’ behavior and can deal with high-dimensional data [8]. Moreover, ML approaches can perform nonlinear relations in data and are somewhat flexible to outliers.

Several studies have focused on developing accurate ML models to reduce downtime in production and improve process quality; most deal with continuous processes. Recently, Kojuk et al. [9,10] developed a method and approach to building a decision support tool combining supervised and semi-supervised techniques to detect and diagnose faults performed over data from continuous processes. Ren et al. [11] developed a methodology based on deep belief networks and multiple models to accomplish fault detection for complex systems. Chiu et al. [12] proposed a method using random forest and a time-series deep-learning model based on the long short-term memory networking to achieve real-time monitoring and faster corrective adjustment of machines. Furukawa et al. [13] used the change score generated by the ChangeFinder as new features at the SVM to classify normal and anomalous conditions improving the detection speed and accuracy compared to the original SVM. Finally, Makridis et al. [14] proposed a method that combines an ensemble to perform the task of predicting faults in maritime vessels.

Regarding FDD tasks in discrete event systems, such as DMMs, most studies represent the system’s behavior by Petri Nets or Finite State Machines to implement the diagnostic approaches. For instance, Cohen et al. [4] developed a hybrid approach that uses Petri Nets to guide data-driven fault diagnosis of PLC (Programmable Logic Controller)-timed cyclic event systems with a 97.2% validation accuracy. Furthermore, Lee and Chuang [15] developed a Petri Net-Based Fault Diagnostic System for Industrial Processes. Their solution involves learning the machine’s normal behavior, designing a Petri Net from it, and implementing PLC routines based on logical combinations that detect if the current machine behavior is normal or anomalous. Finally, Ghosh et al. [5] developed an automated fault detection tool for PLC-controlled manufacturing systems. Their approach is centered on learning the states of a sequential machine over time to detect when a sensor or actuator state change occurs in an unexpected moment.

Therefore, some significant findings may be highlighted from the studies mentioned above. Firstly, research on ML for RT-FDD in DMMs remains lacking since most studies that employ ML to FDD are focused on continuous processes, while those that deal with discrete-event systems, such as DMMs, mainly deal with timed events and use Petri Nets or State Machines. In addition, the combination of continuous variables with discrete events has not been identified in any prior study regarding RT-FDD for DMMs, even though continuous variables may indicate an imminent failure and contribute to the improvement of the FDD task. For instance, an unusual temperature at a specific device may precede its damage. Moreover, the typical sequential cyclic behavior of DMMs has not been considered in any prior study dealing with ML to perform the RT-FDD task.

Thus, considering that ML may significantly contribute to RT-FDD in DMMs, another challenge must be faced: the current industrial workforce barely includes professionals ready to use ML, such as data scientists [16]. With that in mind, Automated Machine Learning (AutoML) has been employed by researchers to address this gap, enabling non-ML experts to explore ML technologies [17,18,19,20].

AutoML is essentially a paradigm related to automating the entire or part of the ML process to reduce human effort on model development and empower domain experts to use machine learning [17]. In this sense, Larocque-Villiers et al. [21] and Li et al. [22] developed AutoMLs for intelligent fault detection on bearings and gearboxes, respectively. Kefalas et al. [23] investigated the usage of AutoML for Remaining Useful Life Estimation of Aircraft Engines, and Nascimento et al. [22] studied the diagnostic of operation conditions and sensors faults using an AutoML. In all cases, the AutoML proved very efficient and saved significant time.

Keeping in mind the improvement of the manufacturing industry performance by reducing downtime with better fault diagnosis, this work proposes a novel and domain-specific AutoML approach for RT-FDD in DMMs. It explores the cyclic sequential behavior of DMMs, considers the scarcity of ML professionals in the industry, and uses only data commonly available in industrial SCADA (Supervisory Control and Data Acquisition) systems: time series of digital and analog IOs. In this sense, the main contributions of this research are:an AutoML approach that enables non-ML experts to implement data-driven RT-FDD in the industry since it requires human contributions only in the automation and maintenance domains;a method to combine discrete events and continuous variables composing the features for RT-FDD in DMMs, that considered its cyclic sequential behavior;the evaluation of how the combination of discrete timed-events and continuous variables as features contributes to the enhancement of models’ performance;the evaluation of the generated models’ capacity to correctly diagnose faults, even when only a few samples of the faulty conditions are available.

The remainder of this document is structured as follows: Section 2.1 (AUTO-ML APPROACH FOR RT-FDD) details the structure of the proposed Auto-ML approach, describing the feature and dataset preparation, the classifiers explored at the model selection mechanism, and the model execution routine; Section 2.4 (3D REAL-TIME MACHINE AND FAULT SIMULATION) describes two simulated machines and nine faulty conditions that are used in the experiments; Section 3 (RESULTS AND DISCUSSION) evaluates the automatically generated models’ capacity to detect and diagnose the faults, and its performance in different scenarios that reflect real-world situations.

## 2. Materials and Methods

### 2.1. Auto-ML Approach for RT-FDD

The proposed approach deals with RT-FDD as a supervised classification problem to identify known faulty situations. It aims to automatically generate a model capable of accurately performing the RT-FDD task and support its re-generation as new situations occur. As shown in Figure 1, it automates all the ML processes and requires the following information, which is limited to the automation and maintenance domains:the initial event of the sequential cycle;the analog IO variables (i.e., temperatures, pressures, distances, positions, and speeds);the digital IO variables (i.e., sensors’ status and actuators’ commands);labels regarding the machine’s working status (i.e., normal, faulty, or anomalous).

The following subsections are dedicated to explaining each process in detail and are summarized in Figure 1.

### 2.2. Automated Model Development

The model development consists of preparing the feature set and the dataset, and selecting the model that best performs among 16 different classifiers. Then, it can be executed to implement the model for the first time and re-create it when new data becomes available.

#### 2.2.1. Process I: Feature Set Preparation

The approach explores two kinds of features: the discrete events during a typical machine cycle and the values of continuous variables when these events occur. The raw data are a time series of each digital and analog IO collected from the PLC. First, discrete events are identified and listed. This process requires a human contribution to specify the first event in the sequential machine cycle and define whether the time series variables are discrete or analog. Then, one new feature is added to each continuous variable for each discrete event in a cycle. Consequently, its outcome is the feature set combining discrete events and continuous variables.

Figure 2 shows a generic example of the feature set assembling proposed. It contemplates:discrete events that occur in a machine cycle;the value of the continuous variables during an interval from the beginning of the cycle;and the generated feature set.

In this example, two events (EV1, EV2) occur at the instants t1 and t2, and the behavior of two analog variables are represented by the red and green dashed lines. So, the feature set contemplates one feature for every event that occurs, EV1_D and EV2_D, and one feature for the values of the continuous variables at the moment the discrete events occur, EV1_V1, EV1_V2, EV2_V1 and EV2_V2.

#### 2.2.2. Process II: Dataset Preparation

As represented in Figure 2, the Dataset is constructed using the data collected from the machine’s PLC data and the feature set generated during the mentioned process I. Each instance refers to one machine cycle and is filled as follows:1.Each n discrete event delay feature (EVn_D) is filled with the time elapsed between its occurrence and the initial cycle event;2.For each n discrete event delay feature (EVn_D), k continuous variables features are filled with their current value when (EVn_D) occurs.3.All missing data are filled with a negative number with -1 since all valid values are positive.

Timed delays of discrete events are pointed to bring evidence of events out of order, delayed, or in advance, and continuous variables enable to detect miscalibration on instruments, malfunction of actuators, dirt, obstruction, or issues on utilities such as abnormal air pressure or electric power.

The human contribution is required for labeling the dataset defining whether the cycles are faulty or non-faulty, together with the specification of the fault. Therefore, examples of labels are N (for non-fault) and Fn (for fault number n). The result of the labeling step is the labeled dataset which is the outcome of Process 2.

#### 2.2.3. Process III: FDD Model Selection

In order to identify the model that best performs the classification task, a total of 16 classifiers are trained and evaluated: Logistic Regression (LR), K Neighbors Classifier (KNN), Naive Bayes (NB), Decision Tree Classifier (DT), SVM—Linear Kernel (SVM), SVM—Radial Kernel (RBFSVM), Multilayer Perceptron (MLP) Classifier, Ridge Classifier (RIDGE), Random Forest Classifier (RF), Quadratic Discriminant Analysis (QDA), Ada Boost Classifier (ADA), Gradient Boosting Classifier (GBC), Linear Discriminant Analysis (LDA), Extra Trees Classifier (ET), Light Gradient Boosting Machine (LIGHTGBM), and Gaussian Process Classifier (GPC).

As represented in Figure 3, 70% of the available dataset is used for training and testing models; and the remaining 30% is reserved as unseen for the final evaluation of all models (unseen dataset). Each one of the 16 classifiers is implemented 30 times [0…29] with different random states to evaluate the classifiers’ sensitivity to the algorithm initialization and train/test split. The Pycaret [24] low-code library is used to automatically perform feature scaling, train/test split, outlier removal, feature selection, hyper-parameters grid search, and model training. The low-code library implements Sklearn [25] classifiers and handles their particularities.

At a later stage, all models are evaluated using the F1 Score so that the influence of True Positives (TP), False Positives (FP), and False Negatives (FN) is considered. The F1 Score distributions over the test and unseen data are separately ranked and compared. Finally, the model that best performs over unseen data is chosen to be applied in Process IV, which regards the RT-FDD task execution.

### 2.3. Model Execution

Once the model is implemented, it can be applied to perform the RT-FDD task even in devices with lower computational power, such as PLC systems. In addition, if new known or novel faults occur in the machine, a new model can be generated by performing the Automated Model Development (Processes I, II, and III) in an adequate robust computer and then deployed again in an embedded system.

#### Process IV: RT-FDD Task Execution

As represented in Figure 1, at this process, a time-series data stream is collected from the PLC and transformed into the unlabeled dataset as established in Process II. As new instances of complete cycles are finished, those are submitted to the classification model chosen to provide the diagnostic in real-time.

The deployment of the task execution can be implemented on edge [1], fog, or cloud, with the data collection system reading the time-series data stream from the PLC. Furthermore, when the automation system includes Linux-based computational modules that support ML model deployment, such as Siemens and Rockwell PLC Systems, the RT-FDD task may be executed even on the device.

At the end of each cycle, the fault detection and diagnosis are elaborated and submitted to model execution when a new instance is generated in the pre-processing task. Therefore, in this study, the real-time term is applied because the detection and diagnosis are provided cycle-by-cycle immediately after the end of each complete cycle.

### 2.4. 3D Real-Time Machine and Fault Simulation

This work uses data generated from simulated machines to evaluate the proposed approach. The simulation is necessary due to insufficient datasets related to industrial production lines [26] and the absence of datasets contemplating time series from real industrial DMMs, with digital and analog IO signals.

The suitability of using data from simulated and real sources for studying fault detection and diagnosis has been investigated by Huan et al. [27]. The authors verified that using both sources is suitable if models are trained and evaluated with data from the same source. In this context, it is worth mentioning that there are other approaches, such as Transfer Learning [28,29,30], that enable applying a model trained with one data source to another one. However, it is not in the scope of this research.

Two real-time simulators, a pick and place system, and an electric furnace machine are used to evaluate the proposed approach’s capacity to generate a suitable model for RT-FDD and its performance. The machine’s choice relies on the fact that they have significantly different dynamic characteristics: the pick and place is a fast discrete positioning system, whereas the furnace is a dynamic system with intrinsic thermal inertial. In addition, both machines are widely present in industries, have sequential and cyclic characteristics, and their automation systems include digital and analog IOs. The simulations were implemented in Unity 3D: a game engine with graphic and physics simulation resources, with the following specifications:The pick and place machine is implemented using forces systems that simulate the power of the motors, as well as frictions and loads;The electric furnace machine is implemented using the dynamic model of an electric heating system and discrete simulation for door conditions.

Images of each machine and their SFCs (Sequential Function Chart) are shown in Figure 4. The simulated faults are related to their primary devices: the heater of the furnace and the linear movement system of the pick and place. Both have been explored to evaluate the generalization capacity of the proposed approach to automatically generate unique models for each machine, capable of distinguishing their normal and faulty conditions.

The data were collected from the simulated machines with a sampling interval of 50 ms and stored in time series format with a timestamp, the variable’s name, and their respective values. Then, the time-series format is converted to the cyclic behavior representation format, as represented in Figure 2.

#### 2.4.1. Pick and Place Machine

The Pick and Place machine is composed of a 3-axis linear positioning system. A PLC controls the movement on each axis with two digital outputs commanding the forward and backward movement, and analog inputs receive the position of each axis with values between 0.0 and 10.0, corresponding to the 0.0% and 100% of the moving range, respectively. Therefore, this system contemplates six digital outputs and three analog inputs.

As represented in Figure 4, the Pick and Place machine is a linear positioning system commanded in three positions in sequence: Position 1 (P1) is [8,9,10], Position 2 (P2) is [5,5,5], and Position 3 (P3) is the origin [0,0,0]. The machine must pick an object at P1, move it to P2 for some intermediary task, and then move it again to P3 to deliver the object. The controller sets the N position at each step by making SET_POS_N=1, and then the equipment moves until the target position is met and POSITION_N_OK=1.

As summarized in Table 1, two kinds of faults are simulated on each of the 3-axis: a punctual obstruction (F1–F3), which emulates punctual damages on a linear guide or fuse, and a speed loss (F4–F6), which emulates a loss of power on the motor driver, a maladjustment, or a lubrication issue that increases the friction of the system. The dataset includes 100 samples of non-fault operation cycles and 100 samples for each simulated faulty situation.

#### 2.4.2. Furnace Machine

The Furnace machine comprises an electric heating system with one on-off and one low-high power command, a temperature sensor, and a door that can be opened and closed, loading and unloading the equipment with metal. The door has sensors to detect its status (open or closed), and there is a button to start the process. Therefore, the Furnace Machine has one analog input, three digital inputs, and four digital outputs.

As shown in the SFC in Figure 4, the operation cycle includes loading the equipment while the door is opened, then closing the door and heating the furnace to three different temperature values (95 °C, 100 °C, and 200 °C) at specific times, and finally returning to the environmental temperature (25 °C). Next, the door is open for unloading, and the cycle is repeated.

Three faulty conditions are simulated: a 2% power loss at the heater (F1), a thermal noise with a magnitude of 5 °C (F2), and a 2 °C spam error (F3). Following the same methodology, the dataset comprises 100 samples of non-fault operation cycles and 100 samples for each of the three simulated faulty situations.

## 3. Results and Discussion

In this section, the results obtained from applying the AutoML approach are presented and discussed. Experiments were performed with two datasets for each simulated machine: one was generated as detailed in Process II of the proposed approach, named PA dataset, and the other with only the discrete timed events, named ODE dataset. The investigations verified:the overall performance of the selected models and the influence of combining discrete and continuous variables (Section 3.1);the performance of all 16 models implemented with different classifiers in the model selection process, their sensitivity to the dataset split, and the initialization (Section 3.2);the performance by class of the selected models using a confusion matrix (Section 3.3);the relevance of timed-events and continuous variables features from a feature importance analysis (Section 3.4);the impact of the dataset size on the performance of the models using the F1 Score (Section 3.5).

### 3.1. Overall Performance Evaluation

The approach was capable of generating models to diagnose faults with F1 Scores of 100% and 85% for the Furnace and Pick And Place simulated machines, respectively. These results were obtained when evaluated with the reserved unseen fraction of the PA dataset and can be observed in Figure 5c and Figure 6c. For the Furnace, the selected model was the Extra Trees Classifier, and for the Pick And Place, it was the Random Forest Classifier.

Figure 5 presents boxplots of all 16 automatically generated classifiers on the model selection process for the Furnace machine experiment, ranked by F1 Score. Figure 5a,b enable a direct comparison between models implemented with PA and ODE test data, respectively. Finally, Figure 5c,d show the performance of the implemented models over reserved unseen data. Following the same organization, Figure 6 presents the same performance analysis for the Pick and Place machine. The red dashed line provides a clear visualization of how models implemented with the PA dataset present a superior mean F1 Score compared with those implemented with the ODE dataset.

Regarding the contribution of continuous variables when combined with discrete events as features, the selected models implemented with the PA dataset present an F1 Score 6% higher than the best models implemented with the ODE dataset. The mean F1 Scores with the ODE and PA datasets were 94% and 100% on the Furnace data, and 79% and 85%, on the Pick And Place data. The difference between the distributions is confirmed by applying a hypothesis t-test with a confidence interval of 95%.

In practical terms, this increment of 6% in the F1-Score should result in higher machine availability due to faster and more precise interventions based on correct diagnostics. The reason is that since the F1 Score strongly considers False Positives and False Negatives, the model is less likely to diagnose a faulty situation when the machine is working in normal condition or vice versa.

No comparison with other studies was performed because, to the best of the authors’ knowledge, there is no prior study nor public benchmark dataset that combines discrete events and continuous variables from a DMM for RT-FDD. The fact is that some of the few publicly available datasets regarding industrial systems are related to continuous process [31,32,33].

### 3.2. Considered Classifiers Performance

It is possible to observe in Figure 5a,c and Figure 6a,c that the two best models, ET and RF for the Furnace, and RF and LIGHTGBM for the Pick And Place, were the same on the evaluations with test and unseen data. In addition, these models presented the highest mean F1 Scores, with the highest minimum values and lowest standard deviations when evaluated with test and unseen data. This low variance suggests models are less sensitive to the train/test split and also to the initialization.

Regarding the performance of the other 14 classifiers, 3 presented F1 Score below 80% on the Furnace dataset, and 8 presented performance below 70% on the Pick And Place dataset. This evidences that not all classifiers learn adequately from data. Moreover, regarding the sensitivity of the classifiers to the initialization and the dataset split, it is possible to verify that some classifiers present a significantly lower sensitivity than others by observing the standard deviation of each distribution. This observation is relevant since a highly sensitive classifier to the initialization or dataset may overfit the data and present an unsatisfactory performance over unseen data.

### 3.3. Performance Evaluation by Class

Table 2, Table 3, Table 4 and Table 5 show how each class on unseen instances are labeled on the ODE and PA datasets. As can be observed, the classifiers implemented with the PA dataset presented an improved performance in all classes. Regarding the improvements on the Furnace RT-FDD model, in Table 2, faults F1 and F3 presented wrong diagnostics. These misclassifications are entirely eliminated when the model is implemented with the PA dataset, as observed in Table 3.

As summarized in Table 4 and Table 5, considering the RT-FDD models for the Pick And Place machine, correct classifications in all classes are improved by applying the PA dataset. In addition, significant improvements are observed on Faults F1 and F4, enhancing more than 50% for correct classifications. Therefore, despite remaining inefficiency regarding some classes, the models implemented with the PA dataset still present significantly superior capacity to provide a correct diagnosis.

### 3.4. Feature Importance Analysis

An ablation study was performed to analyse the importance of each feature, by removing it from the dataset and verifying the model’s performance in its absence. As can be observed in Figure 7 and Figure 8, of the 10 most relevant features, 6 (>50%) are continuous variables within the red dashed rectangles. This result corroborates the positive contribution of the combination of continuous variables and discrete events in the dataset from the evidence that continuous variables features are as relevant as the most relevant discrete event features. This analysis was performed using Pycaret’s model evaluation module.

### 3.5. Sensitivity to the Dataset Size and Improvement Capacity

Since faults are rare events, it is reasonable to expect that an initial dataset in a new deployment of the proposed AutoML approach may contemplate a small number of samples of faulty cycles. Therefore, the model’s sensitivity to the number of faulty cycle samples and its capacity to improve as new known faults occur was investigated. It involved training and evaluating the selected models with datasets that count 20 to 70 instances of each class, in steps of 5 (i.e., [20, 25, …, 70]).

As can be seen in Figure 9 and Figure 10, the models implemented with the PA dataset present superior performance compared to those implemented with the ODE dataset. Furthermore, models implemented with the PA dataset and evaluated over unseen data present a more evident performance enhancement as the number of instances increases compared to those implemented with the ODE dataset.

As can be seen in Figure 9c,d, in the evaluation with unseen data, the F1 Score of Furnace classifiers implemented with the PA dataset shows a gain from 82% to 94% when the number of faulty samples increases from 20 to 70. In the case of the ODE dataset, the F1-Score gain is limited to 85%. Similar behavior is observed for the best classifier of the Pick And Place example. Increasing the samples from 20 to 70, the F1 Score improved from 80% to 88% and from 74% to 82%, with PA and ODE datasets, respectively.

Therefore, the models implemented with PA and ODE datasets are equivalent in correctly diagnosing a small number of fault samples. However, models implemented with the PA dataset are significantly more capable of improving as they are retrained with new fault samples.

Finally, Figure 11 summarizes the distribution of the F1 Score considering all the models implemented over PA and ODE datasets and the dataset sizes from 20 to 70 instances for each machine. As observed, models implemented following the proposed approach (PA) present higher performance than the models implemented only with discrete events (ODE).

## 4. Conclusions

A new automated ML approach for real-time fault detection and diagnosis (RT-FDD) in discrete manufacturing machines (DMMs) is presented and validated with two case studies: a Furnace and a Pick And Place simulated machines. The models generated by the approach presented the highest mean F1 Scores and the lowest variances among all 16 classifiers considered in the model selection process. Extra Trees and Random Forest are the classifiers selected for the Furnace and Pick And Place Machines, with average F1 Scores of 100% and 85%, respectively.

A significant improvement of 6% in the mean F1 Score is verified when continuous and discrete variables are combined following this study’s proposed approach compared with a dataset built with only timed-delay discrete events. The statistical difference in the distributions is confirmed by applying a hypothesis test with a 95% confidence interval. Moreover, in the feature importance analysis, six (6) continuous variables are listed within the ten (10) most relevant features, corroborating the positive contribution of the combination of continuous variables and discrete events in the dataset.

In both case studies, the classifiers implemented with the PA dataset show an F1 Score higher than 80% when only 20 samples of each fault class are available. The F1 Score is enhanced to over 90% when 70 samples of each faulty condition are available. These results show the approach’s capacity to diagnose faults when it is first deployed as well as its capacity to improve each time a new fault occurrence is detected.

Future works should consider the introduction of automatic anomaly detection for identifying novel faulty conditions, clustering, and explainability resources to support understanding and labeling the new cases. Furthermore, new studies that contemplate the deployment of AutoMLs in industry and their impacts may highlight new gaps and research directions to remove barriers that prevent ML from being widely used on the shop floor.

## Figures and Tables

**Figure 1 sensors-22-06138-f001:**
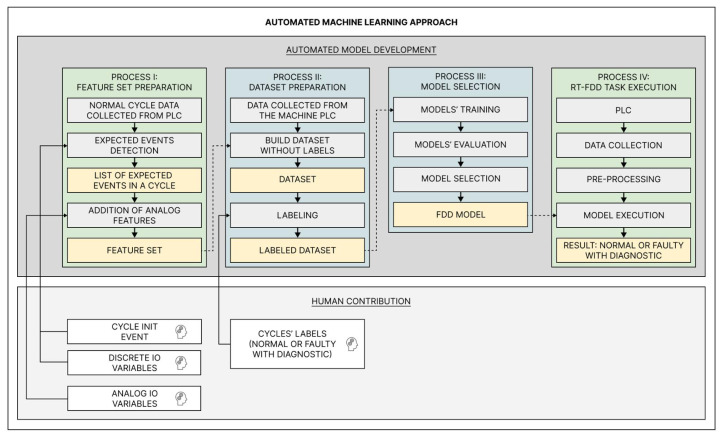
Flowchart of the proposed AutoML approach.

**Figure 2 sensors-22-06138-f002:**
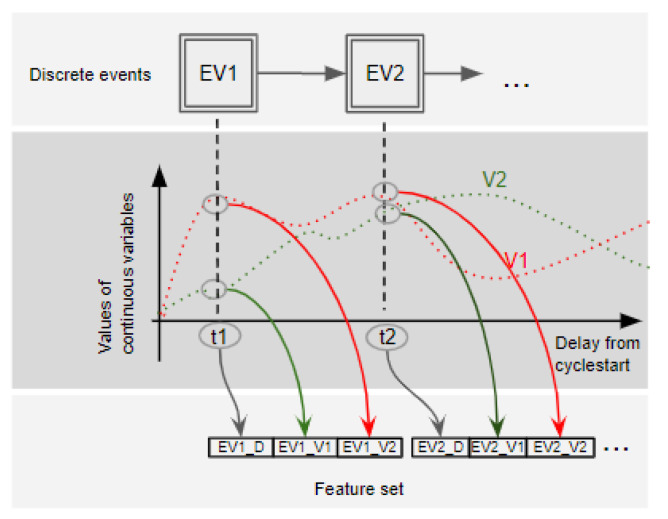
Feature set preparation process to combine discrete events and continuous variables.

**Figure 3 sensors-22-06138-f003:**
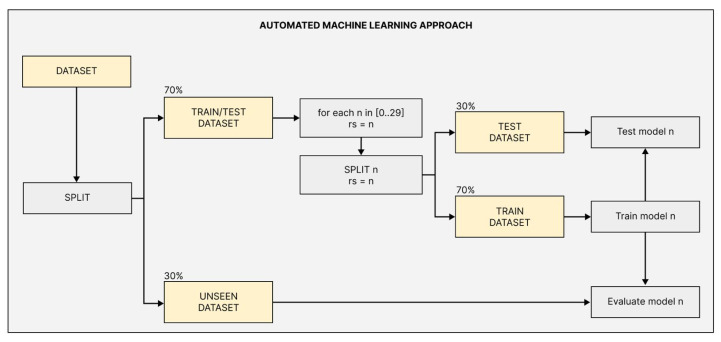
Flowchart of the classifier implementation and evaluation process.

**Figure 4 sensors-22-06138-f004:**
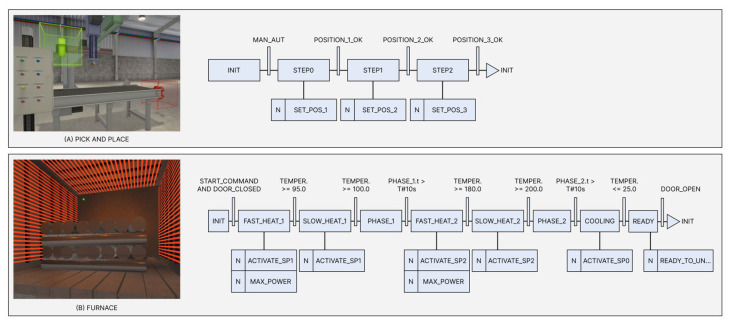
Machine simulators and their SFCs: (**A**) Pick And Place System, with X, Y, and Z movement axis; and (**B**) Furnace, with a heating system.

**Figure 5 sensors-22-06138-f005:**
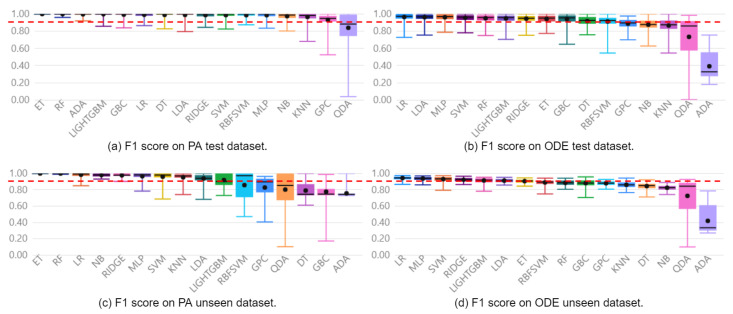
Distribution of F1 Score of Furnace classifiers for Only Discrete Events (**b**,**d**) and the Proposed Approach (**a**,**c**).

**Figure 6 sensors-22-06138-f006:**
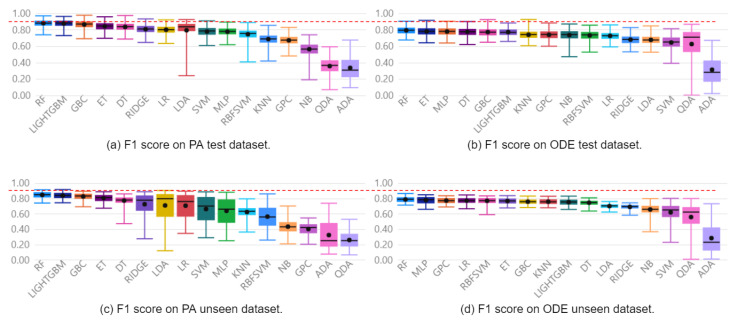
Distribution of F1 Score of Pick And Place classifiers.

**Figure 7 sensors-22-06138-f007:**
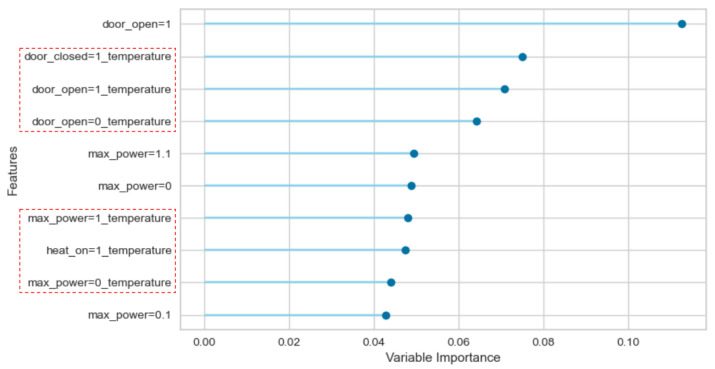
Furnace: Feature Importance regarding the PA dataset.

**Figure 8 sensors-22-06138-f008:**
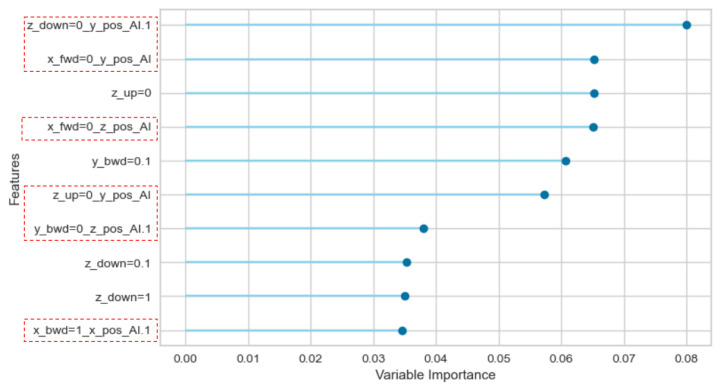
DPick And Place: Feature Importance regarding the PA dataset.

**Figure 9 sensors-22-06138-f009:**
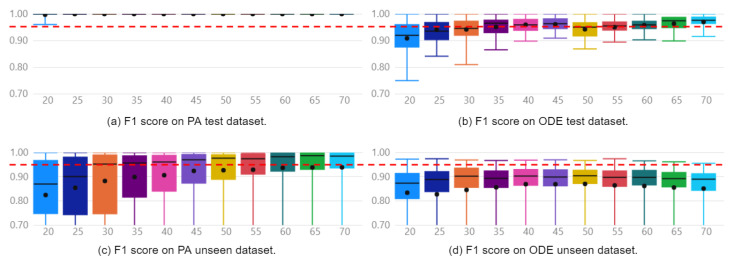
F1 Score of Furnace classifiers by number of instances for training, over: (**a**) PA test data, (**b**) ODE test data, (**c**) PA unseen data, (**d**) ODE unseen data.

**Figure 10 sensors-22-06138-f010:**
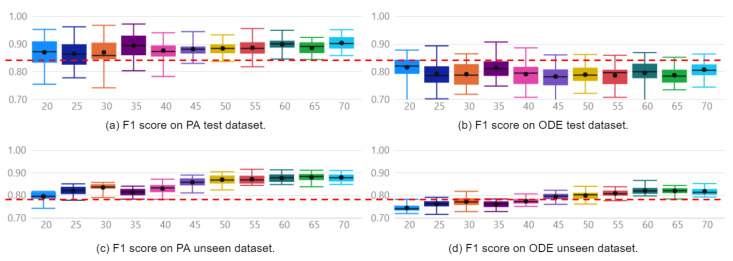
F1 Score of Pick And Place classifiers by number of instances for training, over: (**a**) PA test data, (**b**) ODE test data, (**c**) PA unseen data, (**d**) ODE unseen data.

**Figure 11 sensors-22-06138-f011:**
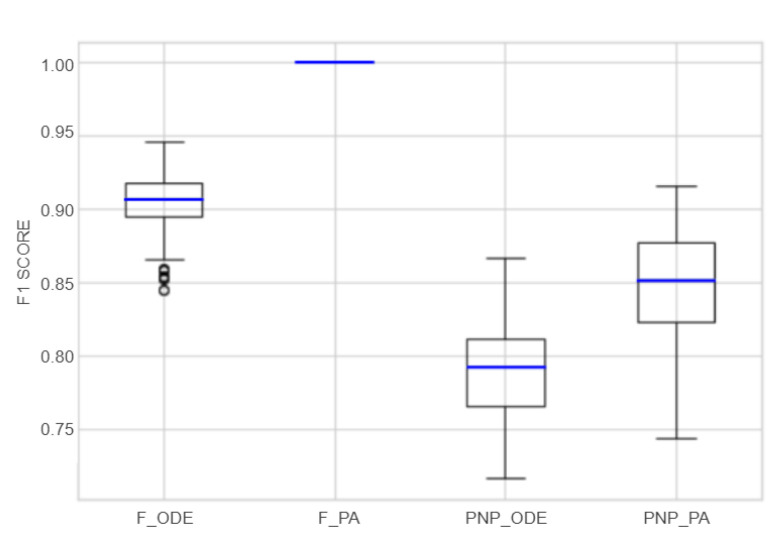
Boxplot with best classifiers for Furnace (F) and Pick And Place (PNP), over ODE and PA datasets.

**Table 1 sensors-22-06138-t001:** Dataset classes and respective numbers of cycles.

Cycles	Class Description
100	Normal operation
100	F1: Punctual obstruction on axis X.
100	F2: Punctual obstruction on axis Y.
100	F3: Punctual obstruction on axis Z.
100	F4: 2% Speed loss on axis X.
100	F5: 2% Speed loss on axis Y.
100	F6: 2% Speed loss on axis Z.

**Table 2 sensors-22-06138-t002:** Furnace: confusion matrix with ODE dataset, over unseen data.

	Predicted				
**Actual**		**N**	**F1**	**F2**	**F3**
	**N**	30	0	0	0
	**F1**	0	23	0	7
	**F2**	0	0	30	0
	**F3**	3	9	0	18

**Table 3 sensors-22-06138-t003:** Furnace: confusion matrix with the PA dataset, over unseen data.

	Predicted				
**Actual**		**N**	**F1**	**F2**	**F3**
	**N**	30	0	0	0
	**F1**	0	30	0	0
	**F2**	0	0	30	0
	**F3**	0	0	0	30

**Table 4 sensors-22-06138-t004:** Pick And Place: confusion matrix with ODE dataset, over unseen data.

	Predicted							
**Actual**		**N**	**F1**	**F2**	**F3**	**F4**	**F5**	**F6**
	**N**	23	5	0	1	0	0	1
	**F1**	11	12	0	0	0	1	0
	**F2**	0	0	29	0	0	1	0
	**F3**	0	1	0	27	1	1	0
	**F4**	0	8	0	1	18	2	1
	**F5**	0	0	1	0	0	29	0
	**F6**	0	0	0	1	0	0	29

**Table 5 sensors-22-06138-t005:** Pick And Place: confusion matrix with PA dataset, over unseen data.

	Predicted							
**Actual**		**N**	**F1**	**F2**	**F3**	**F4**	**F5**	**F6**
	**N**	24	6	0	0	0	0	0
	**F1**	6	19	0	0	5	10	0
	**F2**	0	0	29	0	0	1	0
	**F3**	0	0	0	29	1	1	0
	**F4**	1	1	0	0	28	0	0
	**F5**	0	0	0	0	0	30	0
	**F6**	0	0	0	0	0	0	30

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
