# Peer review of "An Automated Machine Learning Approach for Real-Time Fault Detection and Diagnosis"

_sensors, 2022, doi:10.3390/s22166138_

Round 1

Reviewer 1 Report

This paper proposes AN AUTOMATED MACHINE LEARNING APPROACH FOR REAL-TIME FAULT DETECTION AND DIAGNOSIS. In general, this paper is well presented. The following issues can be further considered.

1. More background and motivation of this study can be added, in case the readers are not very familiar with the topic.

2. The descriptions of the well known knowledge can be properly reduced.

3. Why introducing automated learning method for the fault diagnosis problem? What is the major benefits compared with traditional methods?

4. What is the major benefit of the real time method?

5. Some related works on this topic should be reviewed, such as "Universal Domain Adaptation in Fault Diagnostics with Hybrid Weighted Deep Adversarial Learning", "Federated Transfer Learning for Intelligent Fault Diagnostics Using Deep Adversarial Networks with Data Privacy", etc.

6. A couple of ablation studies should be added to evaluate the effects of the key parameters of the proposed method on the performance.

Author Response

    1. More background and motivation of this study can be added, in case the readers are not very familiar with the topic.

    We agreed with the reviewer that a more elaborated background section would be attractive, especially for readers that are not very familiar with the topic. In this way, we are working on a review article with a background section considering a detailed explanation of the recent research in the area. 

    The introduction section contextualizes different ideas and aspects concerning AutoML and Real-Time Fault Detection and Diagnosis so that readers can identify the article´s novelty. We added a new paragraph (paragraph 2) to the text, that highlights the types of industrial processes, and the differences that demand distinct approaches to deal with RT-FDD in each one of them. Besides, in paragraph 3 we specify that the study focuses on discrete manufacturing machines and present limitations of the current practice used for RT-FDD. 

    “It turns out that industrial processes may be continuous or discrete, and different RT-FDD approaches are required for each one since their behavior are distinct. Continuous processes, such as oil refineries and distillation systems, deliver products at a specific rate (e.g., ton/h, l/min) with uninterrupted operation. While discrete processes, such as Discrete Manufacturing Machines (DMMs), are executed in well-defined sequences of steps, with specific duration, delivering unitary items (e.g., bottles, boxes)[2]. Therefore, performing RT-FDD in continuous processes involves comprehending the normal behavior of variables over time, in steady states and transitions, to differentiate it from anomalous or faulty behaviors. In contrast, on discrete systems, RT-FDD comprises understanding sequential operations, their duration, and how continuous variables behave in each operation.

    In this context, this study focuses on performing RT-FDD in DMMs, which are widely present in the manufacturing industry. Their automation systems generally include knowledge-based (KB) fault diagnostic features that monitor sensors and parameters, and trigger alarms [3]. Furthermore, their behavior is typically described by a sequence of events, represented by inputs and outputs (IOs)[4][5], and may also involve continuous variables such as pressures, flows, temperatures, levels, positions, energy quality, power, or consumption[3]. Consequently, KB fault diagnostic is limited to a few known situations that can be humanly implemented due to the complexity and variety of the data types and the number of devices to be monitored in a DMM. Given that, manufacturing performance may be improved by overcoming the limitations of the KB solutions and delivering more and better diagnostics that support maintenance interventions.”

    Regarding the motivation, we improved the paragraph that states the goal of the research, which is contributing to the improvement of the manufacturing industry.

    “Keeping in mind the improvement of the manufacturing industry performance by reducing downtime with better fault diagnosis, this work proposes a novel and domain-specific AutoML approach for RT-FDD in DMMs. It explores the cyclic sequential behavior of DMMs, considers the scarcity of ML professionals in the industry, and uses only data commonly available in industrial SCADA systems: time series of digital and analog IOs.”

    1. The descriptions of the well known knowledge can be properly reduced.

    From Materials and Methods, we removed the text that mentioned the expected results of the approach (lines 112-116 on the original submitted paper), and combed the remaining 2 paragraphs into one, removing the sentence in lines (118-119 on the original submitted paper).

    The resulting paragraph is:

    “The proposed approach deals with RT-FDD as a supervised classification problem to identify known faulty situations. It aims to automatically generate a model capable of accurately performing the RT-FDD task and support its re-generation as new situations occur. As shown in Figure 1, it automates all the ML processes and requires the following information, which is limited to the automation and maintenance domains:”

    On lines 135-136, the information regarding data types for digital and analog IOs.

    On lines 136-137 the information that discrete events are identified by detecting rising and falling edges from digital IO was removed.

    1. Why introducing automated learning method for the fault diagnosis problem? What are the major benefits compared with traditional methods?

    The major benefit of employing automatic machine learning is to empower non-ML experts in the industry to develop and deploy an ML model for RT-FDD. This is stated in the introduction section in a paragraph and the contribution items:

    "Thus, considering that ML may significantly contribute to RT-FDD in DMMs, another challenge must be faced: the current industrial workforce barely includes professionals ready to use ML, such as data scientists [15]. With that in mind, Automated Machine Learning (AutoML) has been employed by researchers to address this gap, enabling non-ML experts to explore ML technologies [16],[17],[18],[19]."

    In agreement with the Reviewer, we improved the paragraph that states the purpose of the research by highlighting that the new approach considers the scarcity of professionals capable of deploying ML models in industry:

    “Keeping in mind the improvement of the manufacturing industry performance by reducing downtime with better fault diagnosis, this work proposes a novel and domain-specific AutoML approach for RT-FDD in DMMs. It explores the cyclic sequential behavior of DMMs, considers the scarcity of ML professionals in the industry, and uses only data commonly available in industrial SCADA systems: time series of digital and analog IOs”.

    1. What is the major benefit of the real time method?

    Since we are dealing with sequential and cyclic machines, and a fault may not result in an immediate stoppage, a real-time diagnostic will enable early intervention to prevent the stoppage. The first paragraph was improved highlighting the value of the real-time aspect.

    “Industry 4.0 uses IoT, digital twin systems, and predictive maintenance technologies to improve business processes and gain a competitive advantage. In that scenario, Real-Time Fault Detection and Diagnosis (RT-FDD) is fundamental for increasing the reliability of production systems by preventing breakdowns[1]. The real-time aspect enables early interventions when abnormalities are detected, while the diagnosis feature supports precise maintenance actions.”

    1. Some related works on this topic should be reviewed, such as "Universal Domain Adaptation in Fault Diagnostics with Hybrid Weighted Deep Adversarial Learning", "Federated Transfer Learning for Intelligent Fault Diagnostics Using Deep Adversarial Networks with Data Privacy", etc.

    We thank the Reviewer for the suggestion, and we improved the section that presents the dataset creation using simulation by highlighting that transfer learning is also an alternative to overcome the scarcity of data, but is not in the scope of this research. 

    “This work uses data generated from simulated machines to evaluate the proposed approach. The simulation is necessary due to insufficient datasets related to industrial production lines [28] and the absence of datasets contemplating time series from real industrial DMMs, with digital and analog IO signals. 

    The suitability of using data from simulated and real sources for studying fault detection and diagnosis has been investigated by Huan et al. [29]. The authors verified that using both sources is suitable if models are trained and evaluated with data from the same source. In this context, it's worth mentioning that there are other approaches such as Transfer Learning [30][31][32], that enable applying a model trained with one data source to another one. However, it is not in the scope of this research.”

    1. A couple of ablation studies should be added to evaluate the effects of the key parameters of the proposed method on the performance.

    We agreed with the Reviewer on the importance of ablation studies and highlighted this aspect by improving the first paragraph on the subsection that describes the feature importance analysis.

    An ablation study was performed to analyze the importance of each feature, by removing it from the dataset and verifying the model's performance in its absence. As can be observed in Figures 7 and 8, of the 10 most relevant features, 6 (>50%) are continuous variables within the red dashed rectangles. This result corroborates the positive contribution of the combination of continuous variables and discrete events in the dataset from the evidence that continuous variables features are as relevant as the most relevant discrete event features. This analysis was performed using Pycaret's model evaluation module.”

Reviewer 2 Report

The main contribution of this work is to present a novel Automated Machine Learning (AutoML) approach for Real-Time Fault Detection and Diagnosis (RT-FDD), which has important applications in industry. However, the authors should conduct various modifications before the paper can be evaluated for publication.

1.      How to embody the " Automated Machine Learning" of this article? Such as automatic feature selection, automatic data cleaning and lack of data interpolation, automated model selection, and so on. This is very important since this is the novelty of the research.

2.      Line 312-314, “to the best of the authors’ knowledge, there is no prior study nor public benchmark dataset that combines discrete events and continuous variables from a DMM for RT-FDD”. However, accurate tracking and correct diagnosis of failures in complex systems with mixtures of discrete and continuous variables have been studied in

[1] Lerner, Uri, et al. "Bayesian fault detection and diagnosis in dynamic systems." Aaai/iaai. 2000.

3.      How do you deal with the hyperparameters in some classifiers? Such as Decision Tree Classifier, Random Forest Classifier, and so on.

4.      There are a lot of abbreviations in the manuscript, which is not conducive to reading. In general, a word or phrase can be abbreviated only when it appears three or more times in the text. The “DBNs-MMs” in Line 41, “NB” in Line 175, and “RIDGE” in Line 177, et al appear only one time in the full text. There is no need to use abbreviations.

And some abbreviations are repetitive and confusing. Such as “inputs and outputs (IOs)” in Line 25 and “inputs and outputs (IOs)” in Line 86. And the “random forest (RF)” in Line 42, “Random Forest Classifier (RF)” in Line 177, and “Random Forest Classifier (RF)” in Line 299.

Please check the full text to address this issue.

5.      Please insert figure and table AFTER where it is first mentioned in the text, including Figure 5 and Figure 6, Tables 2-5.

And explain the figure where it is mentioned, as in Section 3.1 the description of Figure 5(a)(b)(d) is missing, and so is Figure 6.

6.      The conclusion is redundant, please simplify.

7.      Line 55 and 57, the annotation should be given when the noun “PLC” first appears.

Author Response

1. How to embody the " Automated Machine Learning" of this article? Such as automatic feature selection, automatic data cleaning and lack of data interpolation, automated model selection, and so on. This is very important since this is the novelty of the research.

We agreed with the Reviewer and we improved the subsection that presents Process III. FDD Model Selection, by adding the following content that explains how we dealt with the mentioned aspects. 

"The Pycaret[ 23] low-code library is used to automatically perform feature scaling, train/test split, outlier removal, feature selection, hyper-parameters grid search, and model training. The low-code library implements Sklearn[ 24 ] classifiers and handles their particularities."

  1. Line 312-314, “to the best of the authors’ knowledge, there is no prior study nor public benchmark dataset that combines discrete events and continuous variables from a DMM for RT-FDD”. However, accurate tracking and correct diagnosis of failures in complex systems with mixtures of discrete and continuous variables have been studied in

[1] Lerner, Uri, et al. "Bayesian fault detection and diagnosis in dynamic systems." Aaai/iaai. 2000.

We thank the Reviewer for the recommendation and we improved the refered paragraph by adding a reference to this and another public benchmark dataset for fault detection. However, the recommended study refers to a dynamic continuous process and not a discrete sequential process. The improved paragraph is as follows:

“No comparison with other studies was performed because, to the best of the authors' knowledge, there is no prior study nor public benchmark dataset that combines discrete events and continuous variables from a DMM for RT-FDD. The fact is some of the few available public datasets regarding industrial systems are related to continuous process [33][34].”

  1. How do you deal with the hyperparameters in some classifiers? Such as Decision Tree Classifier, Random Forest Classifier, and so on.

We agreed with the Reviewer and we improved the subsection that presents Process III. FDD Model Selection, by adding the following content that explains how we dealt with the mentioned aspects.

"The Pycaret [23] low-code library is used to automatically perform feature scaling, train/test split, outlier removal, feature selection, hyper-parameters grid search, and model training. The low-code library implements Sklearn[ 24 ] classifiers and handles their particularities."

  1. There are a lot of abbreviations in the manuscript, which is not conducive to reading. In general, a word or phrase can be abbreviated only when it appears three or more times in the text. The “DBNs-MMs” in Line 41, “NB” in Line 175, and “RIDGE” in Line 177, et al appear only one time in the full text. There is no need to use abbreviations.

The abbreviation DBNx-MMs has been removed from line 41. The abbreviations on lines 173-180 were kept because they are used in the box plot charts in figures 5,6,9 and 10.

And some abbreviations are repetitive and confusing. Such as “inputs and outputs (IOs)” in Line 25 and “inputs and outputs (IOs)” in Line 86. And the “random forest (RF)” in Line 42, “Random Forest Classifier (RF)” in Line 177, and “Random Forest Classifier (RF)” in Line 299.

Please check the full text to address this issue.

The repetition of "inputs and outputs" in line 86 has been removed and only the abbreviated for "IOs" was employed, since it was previously defined in line 25. The abbreviation for random forest has been removed in line 42.

In line 177, the abbreviation "RF" is kept for two reasons: it is now the first time the abbreviation is presented in the text, and it is later used in the boxplot charts in figures 5,6,9, and 10.

Moreover, the full text has been carefully checked and the abbreviation (RUL) from Remaining Useful Life was also removed since it wasn't mentioned in other parts of the document.

  1. The conclusion is redundant, please simplify.

We agreed with the reviewer and removed redundant content that was in the first paragraph, within lines 388-391 in the originally submitted article.

  1. Line 55 and 57, the annotation should be given when the noun “PLC” first appears.

An annotation has been added in line 52 where PLC is first cited. The previous annotation in line 57 was removed.

Reviewer 3 Report

Presented article on: “An Automated Machine Learning Approach for Real-time Fault Detection and Diagnosis” is solidly developed. The work has an appropriate structure and layout. The obtained results were substantively discussed.

Although there is no polemic with other researchers (section 3), the authors make it clear that there are no studies on the study of a comparative data set that would combine discrete events and continuous variables from a DMM for RT-FDD. With the above in mind, I recommend this work to be considered for publication in this journal.

Author Response

Thanks a lot for the revision.

Round 2

Reviewer 1 Report

my comments are well addressed. It can be accepted.

Reviewer 2 Report

Thank you for your responses. All my comments have been addressed.